# Image Perturbation-Based Deep Learning for Face Recognition Utilizing Discrete Cosine Transform

**Jaehun Park and Kwangsu Kim \***

College of Computing and Informatics, Sungkyunkwan University, Suwon 16419, Korea; pk9403@skku.edu
\* Correspondence: kim.kwangsu@skku.edu

**Abstract:** Face recognition, including emotion classification and face attribute classification, has seen tremendous progress during the last decade owing to the use of deep learning. Large-scale data collected from numerous users have been the driving force in this growth. However, face images containing the identities of the owner can potentially cause severe privacy leakage if linked to other sensitive biometric information. The novel discrete cosine transform (DCT) coefficient cutting method (DCC) proposed in this study combines DCT and pixelization to protect the privacy of the image. However, privacy is subjective, and it is not guaranteed that the transformed image will preserve privacy. To overcome this, a user study was conducted on whether DCC really preserves privacy. To this end, convolutional neural networks were trained for face recognition and face attribute classification tasks. Our survey and experiments demonstrate that a face recognition deep learning model can be trained with images that most people think preserve privacy at a manageable cost in classification accuracy.

**Keywords:** face recognition; face attribute classification; privacy-preserving deep learning; convolutional neural network (CNN)

## 1. Introduction

Face recognition has been one of the well-known topics in computer vision for a long time. The face is one of the most popular biometrics; as such, face recognition has become an essential tool in our daily lives [1]. Along with the development of deep learning, face recognition has achieved a human-like performance. Deep learning uses the backpropagation algorithm to learn internal parameters and compute the representation in each layer [2]. Large-scale data collected from numerous users have contributed to the rapid development of deep learning.

However, face data contains the identities of individuals, which can be readily linked to other sensitive personal information, such as health data, causing severe privacy leakage. To make matters worse, deep learning is often trained on images without the approval of the person observed in the image [3]. For example, face images in large-scale training datasets, such as social face classification (SFC) [4] and WIDER FACE [5], are collected from social networking services or search engines without explicit consent, which could violate privacy. In addition, information more than just the person's identity can be inferred from the feature representations of face recognition [6,7]. Therefore, extracting sensitive information, such as gender, ethnicity, and health status, without consent is considered a violation of privacy [8]. For this reason, preserving the privacy of face data in deep learning tasks is indispensable in preventing privacy leakage.

There have been numerous studies to preserve privacy in deep learning. Cryptography-based deep learning protects privacy-risk information by encrypting sensitive contents without compromising model accuracy has high computational complexity. Federated learning [9] is designed to train neural networks locally with each client data. Federated learning provides an advantage in privacy over centralized models because the aggregate server only sees trained models. However, cryptographic and federated learning require

a trusted server; otherwise, the attacker can decrypt the ciphertext or restore the original data from gradients [10]. Therefore, in this paper, the focus is on image perturbation-based privacy-preserving methods, which do not require the trust of all parties. Image perturbation methods can be performed during the image distribution phase to transform the image so that the eye cannot recognize the original image. Image pixelization [11], also called mosaicing, can be achieved by dividing the image into a rectangular grid and averaging the pixels within each grid. Blurring [12] removes the image details by convolving the image with the filter function, such as a Gaussian filter or a bilateral filter.

In this paper, a novel image perturbation method is proposed based on the discrete cosine transform coefficient cutting methods (DCC). Our approach is based on pixelization and the discrete cosine transform (DCT) [13]. The DCT expresses a finite sequence of data points as a sum of cosine functions, transforming the image into a DCT coefficient matrix. Most DCT coefficients have a value near zero, and only a few have a relatively large value. The main idea of DCC is that the larger values are more significant in forming an image; thus, cutting the smaller values to conceal the image detail, in the process protecting privacy.

Image perturbation-based privacy-preserving methods, including our method, vary depending on the level of obfuscation. The notion of privacy is subjective, so for the transformed image, someone may consider the image as having preserved privacy, whereas another may not. Figure 1 displays two examples of the proposed DCC. There would be unanimous agreement that Figure 1a is a privacy-preserving transformation; whereas, opinions with respect to Figure 1b are expected to be more subjective. To overcome this problem, a survey was conducted to determine whether the privacy of the transformed image was preserved. To the best of our knowledge, few studies have conducted surveys on whether transformed images preserve privacy. The original image and DCC-transformed image were presented to the participants to determine whether the two images were perceived as having the same identity. The inability to determine whether the two images are the same means that the privacy of the face image has been preserved.

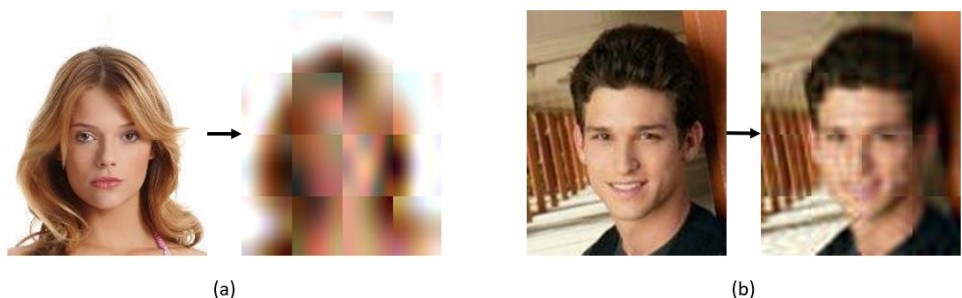

(a)          (b)

**Figure 1.** Overview of our DCC. (**a**) Face image is highly obfuscated by DCC ($a = b = 4$, $r = 32$). (**b**) Face image is weakly obfuscated by DCC ($a = b = 4$, $r = 512$).

Then, face recognition and face attribute classification tasks were conducted. The neural network was trained with DCC-transformed images and tested on the original images. The accuracy dropped by 3–12% depending on the task when trained by the DCC image of ($a = b = 4$, $r = 64$) image, which for most survey participants protected privacy.

Our main contributions are as follows:

1. A privacy-preserving image perturbation method was proposed based on pixelization and discrete cosine transform, that is, DCC.
2. A survey was conducted on whether the proposed method really preserved privacy.
3. A neural network was trained on face recognition tasks with face attribute classification on obfuscated images, achieving satisfactory accuracy, making it suitable for real-world applications.

## 2. Related Works

### 2.1. Convolutional Neural Network

Deep learning processes language, images, audio, and video data mainly using convolutional neural networks (CNNs) [2]. CNN automatically extracts features that distinguish objects from one another, inspired by the classical notion of neurons communicating with other cells via synapses. CNNs have applicability in many domains, such as speech recognition [14], object detection [15], and face recognition [16], and with the development of a large number of datasets recently in new learning algorithms and architecture.

The architecture of CNN includes several bundles of convolution layers and pooling layers, as well as a few fully connected layers. A convolution layer extracts features via the product between each element of the kernel and the input. The output of convolution, called feature maps, is passed through a nonlinear activation function, such as a sigmoid or tanh function, which is a mathematical representation of a biological neuron behavior. A pooling layer downsamples the output to decrease the number of learning parameters. The outputs of the feature maps of the final layer are typically flattened to a one-dimensional array and connected to fully connected layers. The last activation function of the fully connected layer depends on the task of the CNN. In the classification task, of interest is the score of the class probabilities, where each score ranges between 0 and 1, and all scores sum to 1. The training minimizes the loss function through gradient descent and the backpropagation algorithm. The purpose of training is to minimize the difference between the output of the networks and the given ground truth labels.

AlexNet [17] won the challenge in the ImageNet Large Scale Visual Recognition Challenge (ILSVRC) competition in 2012 by correctly classifying ImageNet datasets [18]. The author used the ReLU activation layer [19] to accelerate learning time to improve the network performance. VGGNet [20] uses $3 \times 3$ convolution filters, which push the depth to 16–19 layers, thereby improving the accuracy of the classification of ImageNet. ResNet [21] introduced the skip connection, allowing training with 152 layers while having a lower complexity than VGGNet. As a result, ResNet attained a 3.57% error rate on ImageNet, which is overwhelmingly greater than the human level.

### 2.2. Face Recognition Deep Learning Needs Privacy Preservation

Human faces are often used as training material for deep learning. Initially, faces from images or videos are detected, and their location is determined. After reasonable annotations on the detected face, a deep learning model is trained for face recognition or face analysis.

Face recognition involves identifying or verifying a human in an image. DeepFace [4] trained a network including more than 120 million parameters on four million facial images, with more than 4000 unique identities. An accuracy of 97.35% was attained on the labeled faces in the wild (LFW) dataset [22], overpowering human-level performance in face verification tasks. FaceNet [23] used 100 to 200 million faces with 8 million different identities and achieved an accuracy of 99.63% on face verification tasks using a CNN trained to directly optimize the embedding itself.

Face analysis recognizes valuable information, such as emotion, gender, and age, in images and is utilized for face attribute classification, age estimation, or face mask detection. DTAGN [24] boosts facial expression recognition performance by combining two deep networks: one extracts appearance-related features, and the other extracts geometric features. In [25], a real-time monitoring architecture was proposed to identify face masks using MobileNet V2. In [26], a multitask CNN-based architecture was presented to conduct face analysis tasks concurrently.

Despite the usefulness of face recognition and face analysis, some privacy violation issues have been raised. In [27], it is argued that biometric data can be used to identify a person easily, so in certain cases, malicious leakage can lead to criminality, such as identity theft or tracking of individuals. Therefore, a privacy-preserving mechanism is essential when using biometric data, such as face images.

*2.3. Privacy-Preserving Deep Learning*

Previous studies have been conducted to preserve privacy in the field of deep learning. In [28], a secure face verification system was proposed based on a CNN representation with the Paillier algorithm, saving all the feature representations in ciphertext so that the client would know only the verification result, ensuring privacy. In [29], a novel system was suggested that utilizes additive homomorphic encryption to protect the gradient. However, cryptographic-based methodologies incur a high computational cost.

In [30], a new deep learning algorithm was developed to train a centralized CNN with differential privacy [31], which resulted in decreased accuracy. However, such a centralized CNN needs an honest server because all of the data are stored on the central server. Federated learning has been known to protect privacy, as the central server can only see the local training results while data remain local. Recent studies [10,32] have reconstructed the victim's private data by assuming a malicious server in the federated learning environment. Therefore, federated learning requires a trustworthy server.

Pix [33] extends differential privacy to image data using image pixelization methods; it was demonstrated that Pix can prevent re-identification attacks. PEEP [3] perturbs eigenfaces by utilizing differential privacy to recognize faces. The third-party server only sees the controlled information and consequently preserves privacy. Image perturbation methods, such as Pix and PEEP, do not need to be trusted by third parties because the transformation can be applied at the image distribution stage.

## 3. Methods

This section outlines the entire process of our DCT coefficient cutting method (DCC): Step 1 depicts a formal discrete cosine transform (DCT) and pixelization, Step 2 provides the details of the coefficient cutting method, and Step 3 describes inverse-DCT and presents the results of DCC applied to facial images.

*3.1. (Step 1) Discrete Cosine Transform (DCT)*

The discrete cosine transform (DCT) was first proposed by Ahmed [13] in 1972. DCT transforms a signal or image from the spatial domain to the frequency domain and vice versa for inverse-DCT. One-dimensional DCT (1D-DCT) is used in signal processing [34], and two-dimensional DCT (2D-DCT) is used in image processing [35]. In this study, 2D-DCT was used to transform images. For convenience, 2D-DCT is referred to as DCT in the remainder of this paper. DCT can be applied to both gray and color images. For the color image, DCT is performed in each RGB channel. Let $V$ be the frequency domain, and $X$ be the spatial domain (image). The DCT of an $M \times N$ matrix $X$ is defined as:

$$V_{pq} = \alpha_p \alpha_q \sum_{m=0}^{M-1} \sum_{N=0}^{N-1} \cos \frac{\pi(2m+1)p}{2M} \cos \frac{\pi(2n+1)q}{2N} \qquad \begin{cases} 0 \leq p \leq M-1 \\ 0 \leq q \leq N-1 \end{cases} \tag{1}$$

$$\alpha_p = \begin{cases} \frac{1}{\sqrt{M}}, & \text{if } p = 0 \\ \sqrt{\frac{2}{M}}, & \text{if } 1 \leq p \leq M-1 \end{cases} \qquad \alpha_q = \begin{cases} \frac{1}{\sqrt{N}}, & \text{if } q = 0 \\ \sqrt{\frac{2}{N}}, & \text{if } 1 \leq q \leq N-1 \end{cases} \tag{2}$$

DCT transforms the image of Figure 2a to the frequency domain generating the DCT coefficient matrix of Figure 2b. In the image resulting from the transformation, the white pixels are concentrated at the top left. The whiter the pixel, the larger the DCT coefficient, and the blacker the pixel, the smaller the DCT coefficient. Note that the DCT coefficient values are absolute values. The larger DCT values associated with the lower frequencies represent an essential part of the original image in the transformation back to the spatial domain. This is because the human eye tends to sense the low-frequency components in the picture better. In summary, the most visually important information is concentrated in only a few coefficients of the DCT at the top left. In this study, the image is split into $a \times b$ blocks. As shown in Figure 3b, the DCT is performed blockwise, so it works like a pixelization method.

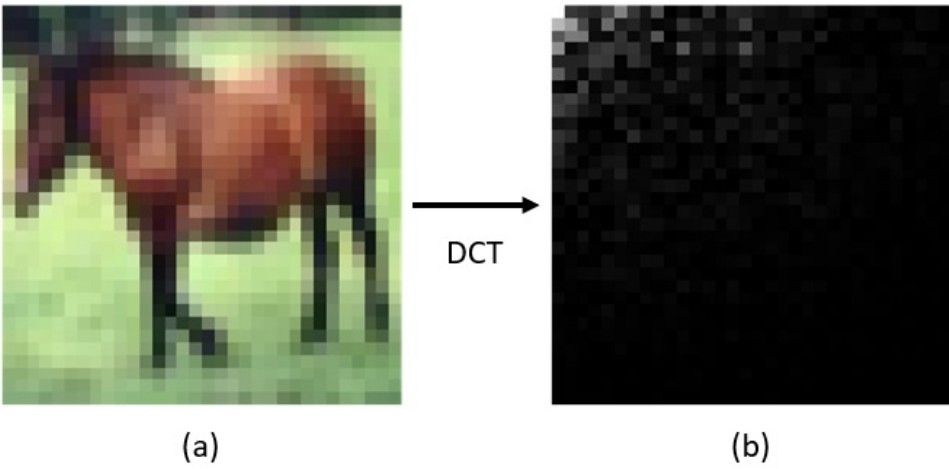

**Figure 2.** (**a**) Sample 32 × 32 image from CIFAR-10 [36]. DCT transforms the image pixel to the frequency domain. (**b**) 32 × 32 DCT coefficient matrix. White pixels represent the maximum DCT coefficient values, and black pixels represent the minimum DCT coefficient values close to zero.

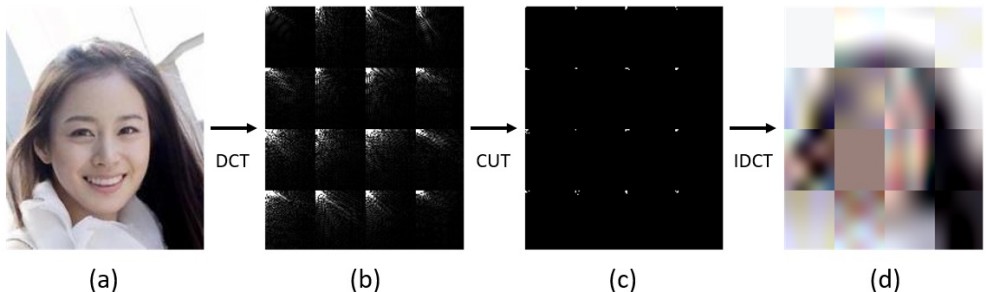

**Figure 3.** Three steps of the DCC process with a sample image from CelebA [37]. (**a**) Step 1 (DCT): The original image is divided into several blocks, and each block is transformed from the spatial domain to the frequency domain by DCT and displayed as the DCT coefficient matrix. (**b**) Step 2 (CUT): The DCT coefficient matrix is filtered by coefficient cutting methods, such that only a few high-frequency coefficients in the DCC coefficient matrix remain. (**c**) Step 3 (IDCT): The DCC coefficient matrix is transformed from the frequency domain to the spatial domain by I-DCT per block. (**d**) Then, the privacy-preserved image is created.

### 3.2. (Step 2) Coefficient Cutting (CUT)

As described in Section 3.1, during the formation of an image, significant information is concentrated at a few low frequencies. The main idea of coefficient cutting is that even if most of the high frequency is omitted, the main features of the image remain intact while the sensitive information is concealed. The largest DCT coefficient for each block was selected and stored in the DCC coefficient matrix, as shown in Figure 4b to maintain at least one DCT coefficient for each block. Except for the selected $a \times b$ DCT coefficients, the top $(r - a \times b)$ DCT coefficients were selected for the whole image, not each block, and stored in the DCC coefficient matrix, as shown in Figure 4c. Then, the remaining coefficients were discarded. The value $r$ is the number of remaining DCT coefficients, which control privacy intensity, and cannot be less than $a \times b$. The larger the $r$ value, the lower the privacy intensity. The coefficient cutting methods filter the DCT coefficient matrix $V$ in Figure 4a, and the DCC Coefficient matrix $V^*$ of Figure 4c is produced as a result.

| 995 | -85 | 25 | 0 | 773 | 212 | 56 | 9 |
|-----|-----|-----|-----|-----|-----|-----|-----|
| 115 | 66 | 77 | 1 | -98 | 125 | 26 | 17 |
| 21 | 51 | -39 | -2 | 15 | 21 | 0 | -23 |
| 5 | -7 | -15 | 3 | 0 | 4 | -7 | 2 |
| 875 | 78 | 52 | -23 | 775 | 221 | 61 | 7 |
| -99 | 22 | -35 | 16 | 121 | -81 | -25 | -12 |
| 41 | 30 | 7 | 4 | -44 | 33 | 18 | 9 |
| -29 | 0 | 12 | 0 | 5 | 7 | 0 | -4 |

(a) DCT Coefficient Matrix V

| 995 | 0 | 0 | 0 | 773 | 0 | 0 | 0 |
|-----|-----|-----|-----|-----|-----|-----|-----|
| 0 | 0 | 0 | 0 | 0 | 0 | 0 | 0 |
| 0 | 0 | 0 | 0 | 0 | 0 | 0 | 0 |
| 0 | 0 | 0 | 0 | 0 | 0 | 0 | 0 |
| 875 | 0 | 0 | 0 | 775 | 0 | 0 | 0 |
| 0 | 0 | 0 | 0 | 0 | 0 | 0 | 0 |
| 0 | 0 | 0 | 0 | 0 | 0 | 0 | 0 |
| 0 | 0 | 0 | 0 | 0 | 0 | 0 | 0 |

(b) Choose Biggest for Each Block

| 995 | -85 | 0 | 0 | 773 | 212 | 0 | 0 |
|-----|-----|-----|-----|-----|-----|-----|-----|
| 115 | 0 | 0 | 0 | -98 | 125 | 0 | 0 |
| 0 | 0 | 0 | 0 | 0 | 0 | 0 | 0 |
| 0 | 0 | 0 | 0 | 0 | 0 | 0 | 0 |
| 875 | 0 | 0 | 0 | 775 | 221 | 0 | 0 |
| -99 | 0 | 0 | 0 | 121 | 0 | 0 | 0 |
| 0 | 0 | 0 | 0 | 0 | 0 | 0 | 0 |
| 0 | 0 | 0 | 0 | 0 | 0 | 0 | 0 |

(c) DCC Coefficient Matrix V*

**Figure 4.** Example of coefficient cutting for $a = b = 2$, $m = n = 8$, $r = 12$. (**a**) DCT coefficient matrix after step 1. (**b**) Select the largest DCT coefficient for each block. (**c**) Select the remaining top $(r - a \times b)$ coefficients. Then, generate the DCC coefficient matrix.

### 3.3. (Step 3) Inverse Discrete Cosine Transform (I-DCT)

In Section 3.2, the DCT coefficients are cut to produce the DCC coefficient matrix $V^*$, which is still in the frequency domain. The inverse discrete cosine transform (I-DCT) changes the frequency domain into the spatial domain. I-DCT is defined as follows: Note that $\alpha_p$ and $\alpha_q$ are the same as in Equation (2).

$$X_{mn} = \sum_{p=0}^{M-1} \sum_{q=0}^{N-1} \alpha_p \alpha_q V_{pq} \cos \frac{\pi(2m+1)p}{2M} \cos \frac{\pi(2n+1)q}{2N} \qquad \begin{cases} 0 \le m \le M-1 \\ 0 \le n \le N-1 \end{cases} \qquad (3)$$

I-DCT transforms the DCC Coefficients $V^*$ to the privacy image $X^*$ of Figure 3d. If the cutting phase is excluded, I-DCT transforms the DCT coefficients $V$ to the original image $X$. Algorithm 1 shows the steps for transforming the DCC images. Figure 5 displays a DCC example with $a = b = 4$. As observed in the figure, DCC hides the personal identity of the person in the image. Figure 5b is equivalent to image pixelization with $r = a \times b = 16$, such that privacy is more strongly maintained. As $r$ increases, the image becomes more comprehensible. Figure 5b is expressed by a DCT coefficient of only 0.04%, and in Figure 5g is expressed by a DCT coefficient of 1.32% within an image size of $178 \times 218$.

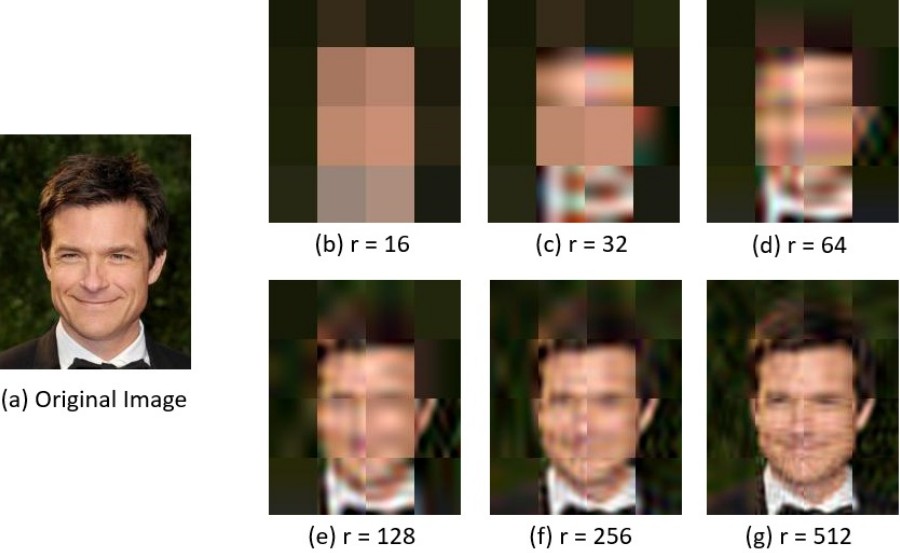

(a) Original Image

(b) r = 16

(c) r = 32

(d) r = 64

(e) r = 128

(f) r = 256

(g) r = 512

**Figure 5.** (**a**) Original image (**b**–**g**) DCC result of a sample image with $a = b = 4$ and different $r$ values. As $r$ increases, the image becomes clearer and it is easier to distinguish who he is.

However, the extent of cutting to apply to the DCT coefficient cannot be determined from the perspective of preserving privacy. Does the image in Figure 5b preserve privacy? The answer to that question is affirmative. However, does the image in Figure 5g preserve privacy? The response is more ambivalent. Therefore, a survey was conducted to understand people's perception of privacy, as discussed in Section 4.

---

**Algorithm 1** Discrete Cosine Transform Coefficient Cutting Methods

---

**Input:** Image, number of blocks $a \times b$, number of remaining coefficients $r$

    Divide an image into $a \times b$ blocks                         ▷ Step 1
    **for all** blocks **do**
        DCT to the image block
        Store largest DCT coefficient value to DCC             ▷ Step 2
    **end for**
    **if** $r > a \times b$ **then**
        Store top $(r - a \times b)$ DCT coefficient values to DCC
    **end if**
    **for all** blocks **do**
        I-DCT to the DCC                             ▷ Step 3
    **end for**
    Combine each blocks
    **return** Privacy-Preserved Image

---

## 4. User Study

This section summarizes the results of the study on people's thoughts towards privacy. As discussed earlier, the notion of privacy is subjective. For example, in the modified image, someone may think that the image still contains sensitive private content, but someone else may think that the image has successfully eliminated private content. The proper degree of DCT coefficient pruning required to protect privacy is vague, so a user study was conducted to explore this issue. For convenience, it is assumed that $a = b = 4$ for the rest of the paper, without any further reference to this notation.

As shown in Figure 6, the survey consists of questions with regard to two images. A celebrity of cultural background similar to that of the participants is considered for the images. Each question refers to two celebrity images of the same or different identities. The image size is $178 \times 218$ pixels, and the face is at the center. The first image is the original facial image without any manipulation. The second image was mutated into a DCC-applied image. Each question consisting of sub-questions was on the same first image and a second image with different privacy-preservation levels. The question asked whether the two images were of the "same person", "different person", or "cannot judge". The first sub-question was on the DCC ($r = 16$) image. The $r$ of DCC was doubled for the next sub-question, which means that the privacy level was lowered. Figure 6a corresponds to the third sub-question, and Figure 6b to the seventh sub-question, which is the lowest privacy level in our survey. The survey respondent could see the next sub-question after answering the current question to prevent cheating. To judge the predictability of the answer, the privacy level of the images was gradually weakened. The main idea of the survey is that failure to determine whether the pair of images are identical implies that the privacy of the face image has been preserved.

There were 69 users in the study, including four pairs of celebrities, and each question consisted of seven sub-questions with respect to DCC transformed images of varying $r$. Therefore, a total of 28 questions were asked. Figure 7 shows that over 96% of people could not judge the identities in the DCC images ranging from $r = 16$ to $r = 64$. This result means that if $r$ is 64 or less, the privacy of the face image is almost preserved. The participants started to correctly identify from $r = 128$ onwards. The number of correct answers exceeded the number of undecided answers for $r = 256$. All participants in the experiment expressed an opinion for $r = 1024$, and most of the participants (97%) answered correctly. This result means that for $r$ greater than 1024, the privacy of the face image is hardly preserved.

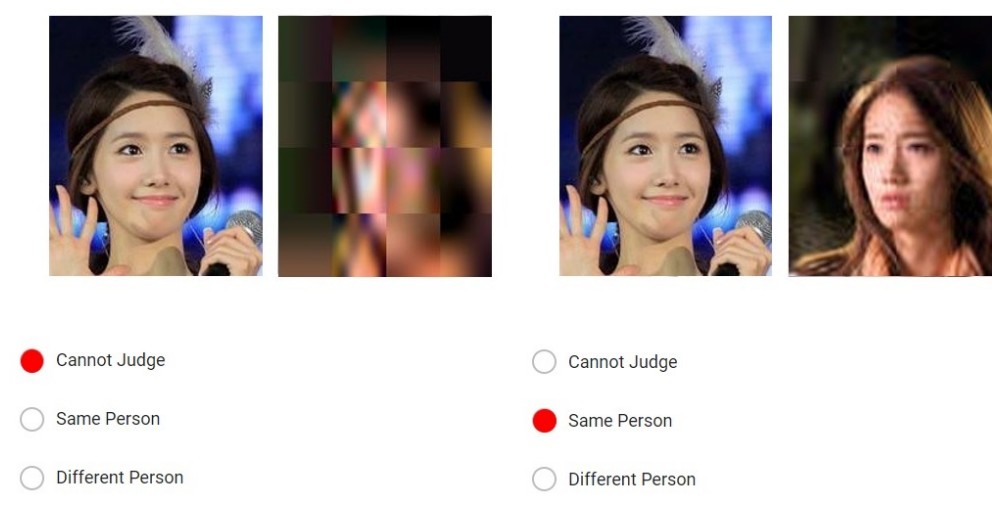

(a) Third sub question                    (b) Seventh sub question

**Figure 6.** The composition of a user study. The image on the left of each sub-question is the original image. (**a**) The image on the right of the third sub-question is transformed by DCC ($r = 64$). (**b**) The image on the right of the seventh sub-question is transformed by DCC ($r = 1024$).

With pixelization, DCC ($r = 16$) preserved privacy the most strongly as the information on the block is compressed into one number. In other words, the pixelated image had only 16 pixels. However, pixelated images are inappropriate for deep learning tasks because of the lack of pixel information. In Section 5, the results are presented on a deep learning experiment conducted to classify face attributes and to evaluate whether face attributes can be recognized by our method.

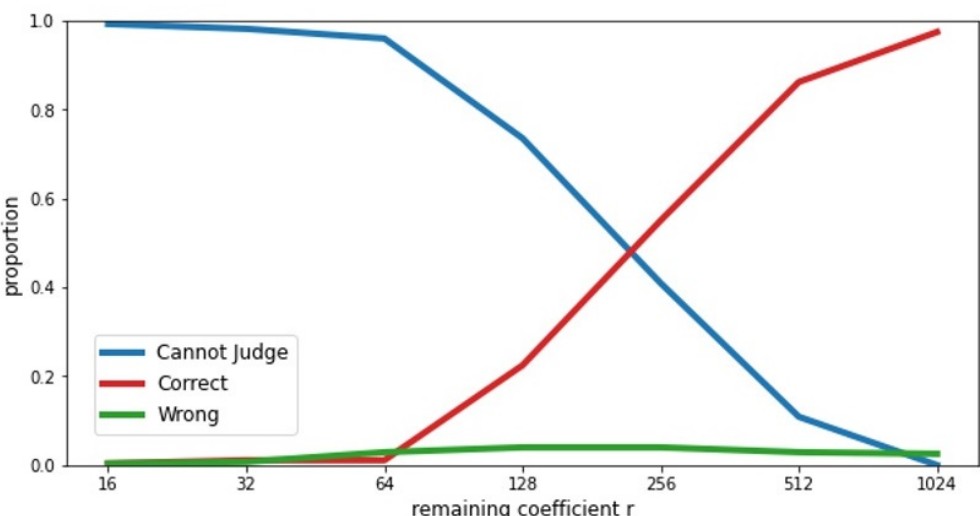

**Figure 7.** Survey Results. Most people could not judge until DCC ($r = 64$).

## 5. Experiment and Results

In this section, two experiments are discussed: face recognition and face attribute classification. As illustrated in Figure 8, the VGG network was modified and the following CNN structures were constructed: 2conv(3 × 3, 64) − maxpool(3 × 3) − 2conv(3 × 3, 128) − maxpool(3 × 3) − 2conv(3 × 3, 256) − maxpool(3 × 3) − 2conv(3 × 3, 512) − maxpool(3 × 3) − flatten() − 2dense(1024) − softmax. The total number of parameters of the CNN is approximately 7.8 M. Batch normalization and ReLu were employed after each convolutional layer. The softmax layer varied depending on the number of classes to predict; an SGD optimizer was used with a learning rate of 0.001, decay of 0.001, and momentum of 0.9; 50 epochs were trained for face recognition and face attribute classification for each

DCC method. As shown in Figure 9, *r* began at 16 and doubled for the next step. The results were evaluated for accuracy on validation data.

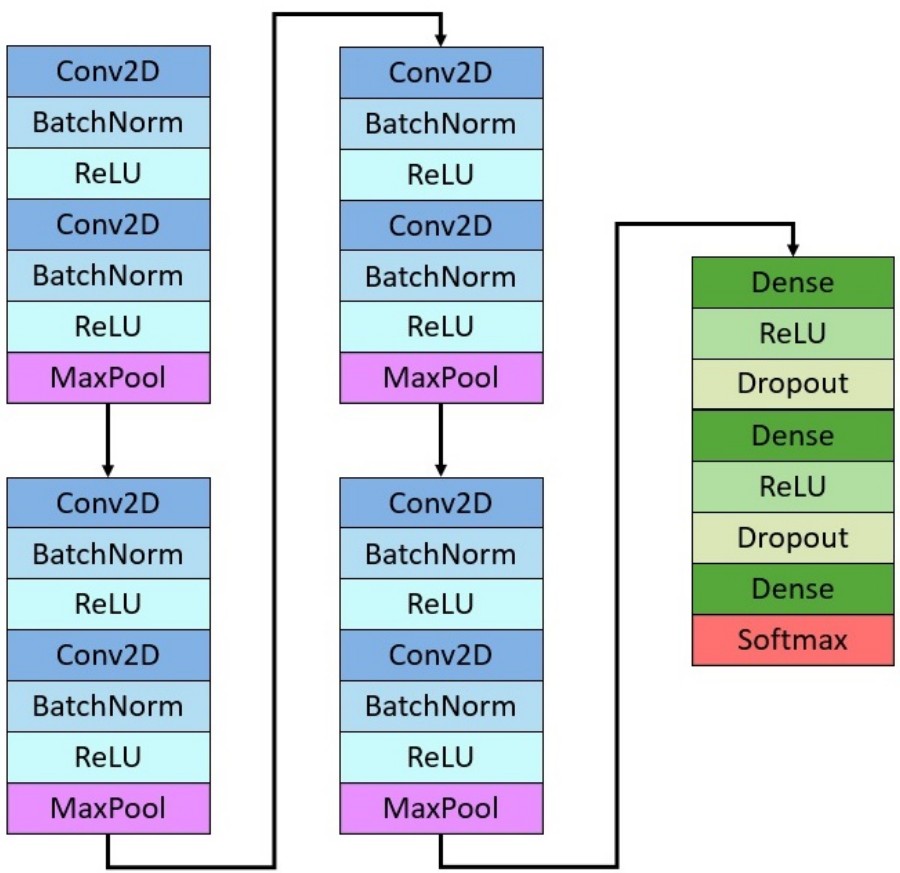

**Figure 8.** Convolutional Neural Network Architecture.

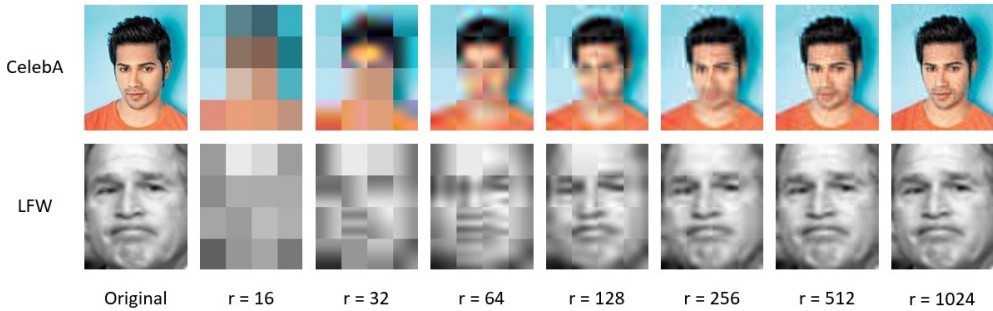

**Figure 9.** Sample image of the training set and variation of DCC with *r*.

### 5.1. Face Recognition

The LFW dataset [22], named "Labeled Faces in the Wild", was used for the face recognition task. The dataset consists of 13,233 black and white images of 5749 individuals, of which only 1140 were used, limiting the minimum number of faces per person to 100, similar to the methods used in [3]. The 1140 images comprised 236 of "Colin Powell", 121 of "Donald Rumsfeld", 530 of "George W Bush", 109 of "Gerhard Schroeder", and 144 of "Tony Blair". Training data and test data were divided to preserve the ratio of samples for each class to prevent imbalance. We used 75% of the input dataset for training and 25% for testing. The class "George W Bush" is overwhelming compared to other classes, so a data augmentation was performed with horizontal flip on all classes except that class. Note that the data augmentation was performed only on the training data. The total number of datasets was 1592 after the augmentation. In addition, five-fold validation was performed

to overcome the shortcomings resulting from a lack of data. The image to be used as input to the CNN was resized to $178 \times 218$.

Table 1 shows the results of face recognition. The accuracy was evaluated using precision, recall, and F1 score. According to the table, the F1 score was approximately 92% when training was performed on the origin data. The F1 score decreased by 49% when the CNN was trained using DCC ($r = 16$). Considering that there are five classes, which means that the initial accuracy is 20%, CNN was rarely learned. However, in DCC ($r = 64$), the F1 score was approximately 82%, which is 10% lower than that of the original one, while maintaining privacy. For $r = 256$ or more, the F1 score was almost the same as that of the original. In conclusion, DCC ($r = 64$) is sufficient for face recognition with privacy protection at the expense of a slight loss in performance.

**Table 1.** Face Recognition Performance.

| | Origin | $r = 16$ | $r = 32$ | $r = 64$ | $r = 128$ | $r = 256$ | $r = 512$ | $r = 1024$ |
|---|---|---|---|---|---|---|---|---|
| Precision | 0.9254 | 0.5398 (−0.3856) | 0.6963 (−0.2291) | 0.8368 (−0.0886) | 0.8870 (−0.0385) | 0.9172 (−0.0082) | 0.9279 (0.0025) | 0.9222 (−0.0033) |
| Recall | 0.9254 | 0.4877 (−0.4377) | 0.6360 (−0.2895) | 0.8316 (−0.0939) | 0.8825 (−0.0430) | 0.9140 (−0.0114) | 0.9254 (-) | 0.9219 (−0.0035) |
| F1 Score | 0.9236 | 0.4298 (−0.4939) | 0.5840 (−0.3396) | 0.8215 (−0.1021) | 0.8759 (−0.0477) | 0.9122 (−0.0115) | 0.9237 (-) | 0.9203 (−0.0033) |

*5.2. Face Attribute Classification*

We used the Large-scale CelebFaces Attributes (CelebA) dataset [37] for face attribute classification. The CelebA dataset consists of more than 200 K celebrity images, each with 40 attribute annotations, such as wearing eyeglasses, an oval face, and wearing a hat or mustache. For each attribute, the value 1 represents having that attribute, and −1 represents not having that attribute.

Four attributes were selected for the experiment: male, brown hair, heavy makeup, and smiling. For each attribute, 10,000 images were extracted, of which 5000 had the attribute, and the remaining 5000 did not; 8000 were used for the training set, and 2000 for the test set, divided by the same ratio. The input size of the image was $178 \times 218$, which is the same as that of the CelebA dataset. Image transformation proceeded in the same way as in the face recognition experiment. The above training procedure was repeated five times with different extraction seeds, which means that the extracted datasets had different configurations. The mean of the accuracy was then calculated and compared with the results.

Table 2 displays the results of the face attribute classification. The accuracy was different for each attribute; that is, male classification had 96% accuracy, and brown hair had 80% accuracy. The number of classes was two, positive or negative, which means that the initial accuracy was 50%. In DCC ($r = 16$), the smiling classification accuracy was 57%, and the male classification was 78%. However, in DCC ($r = 64$), the reduction in the brown hair classification accuracy was 4%, and the smiling classification was 12%, indicating that it could tolerate a decrease in the accuracy while rigorously maintaining privacy. In conclusion, similar to face recognition, DCC ($r = 64$) is sufficient to classify face attributes while protecting privacy, although there is a slight loss of accuracy. If the deep learning service provider requests more accuracy, then DCC ($r = 128$) can be used with weakened privacy preservation and with a reduction in accuracy of 3–4%.

**Table 2.** Face Attribute Classification Accuracy

| Attribute | Origin | $r = 16$ | $r = 32$ | $r = 64$ | $r = 128$ | $r = 256$ | $r = 512$ | $r = 1024$ |
|---|---|---|---|---|---|---|---|---|
| Male | 0.9605 | 0.7776 (−0.1829) | 0.8519 (−0.1086) | 0.8972 (−0.0633) | 0.9308 (−0.0297) | 0.9494 (−0.0111) | 0.9538 (−0.0067) | 0.9574 (−0.0031) |
| Brown Hair | 0.8053 | 0.6655 (−0.1398) | 0.7294 (−0.0759) | 0.7639 (−0.0414) | 0.7743 (−0.0310) | 0.7833 (−0.0220) | 0.7903 (−0.0150) | 0.7990 (−0.0063) |
| Heavy Makeup | 0.9021 | 0.7294 (−0.1727) | 0.7901 (−0.1120) | 0.8308 (−0.0713) | 0.8661 (−0.0360) | 0.8886 (−0.0135) | 0.8897 (−0.0124) | 0.8986 (−0.0035) |
| Smiling | 0.9035 | 0.5747 (−0.3288) | 0.7010 (−0.2025) | 0.7875 (−0.1160) | 0.8584 (−0.0451) | 0.8818 (−0.0217) | 0.8911 (−0.0124) | 0.9063 (0.0028) |

## 6. Conclusions

In this paper, a novel image perturbation mechanism called DCC was proposed to preserve the privacy of face images by combining the discrete cosine transform and pixelization. In addition, a survey was conducted to determine whether the proposed method was able to effectively preserve privacy. Subsequently, the neural networks were trained on DCC-transformed images and tested on the original images. Deep learning results indicated that the DCC can recognize the face and classify face attributes effectively while maintaining privacy.

**Author Contributions:** Conceptualization, J.P. and K.K.; Data curation, J.P.; Formal analysis, J.P.; Funding acquisition, K.K.; Investigation, J.P.; Methodology, J.P. and K.K.; Project administration, K.K.; Resources, J.P.; Software, J.P.; Supervision, K.K.; Validation, J.P.; Visualization, J.P.; Writing—original draft, J.P.; Writing—review and editing, J.P. and K.K. All authors have read and agreed to the published version of the manuscript.

**Funding:** This research was supported by Healthcare AI Convergence Research & Development Program through the National IT Industry Promotion Agency of Korea(NIPA) funded by the Ministry of Science and ICT. (No.1711120339).

**Data Availability Statement:** The data presented in this study are openly available in LFW dataset [22] at http://vis-www.cs.umass.edu/lfw/ (accessed on 19 December 2021) and CelebA [37] at https://mmlab.ie.cuhk.edu.hk/projects/CelebA.html (accessed on 19 December 2021).

**Acknowledgments:** We thank all the 69 people who participated in the survey.

**Conflicts of Interest:** The authors declare no conflict of interest.

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
