# Peer review of "Image Perturbation-Based Deep Learning for Face Recognition Utilizing Discrete Cosine Transform"

_electronics, doi:10.3390/electronics11010025_

Round 1
Reviewer 1 Report
In Figure 2 it looks like DCT has been applied on non-square blocks. DCT is normally used for square matrices, so please correct this figure!
The caption of Figure 7 contains a spelling error (Servey - Survey).
In Section 5.1 is described a rather basic data augmentation technique - using only horizontal flip. Why not scaling, rotation or gamma correction?
Classification methods are evaluated using other parameters also, not only accuracy as in Section 5.1. Confusion matrices, recall, f1 score, etc. can be used. Please add more evaluation parameters and comment on them!
This method is not actually compared to existing algorithms for face recognition with privacy preserving! The overall performance level is not clear.
Author Response
We would like to thank Reviewer 1 for the constructive comments and suggestions for improving the quality of the manuscript. We have carefully reviewed these comments point by point.

Reviewer 2 Report
The paper is well written and the contribution is sound. The study is well designed and the results are satisfactory given the challenge of preserving privacy. Well done.
Author Response
We would like to thank Reviewer 2 for carefully reviewing our papers. Your encouragement was a great help to us. Thank you once again.
Reviewer 3 Report
Below are my questions and comments.
- Image details do not necessarily contain privacy information. Hence, reducing image details may not always hide the sensitive information in a face image.
- How to define sensitive factors in a face image?
- The definition of privacy preserving is confusing. For instance, in a face recognition application, if the images fail to be recognized as the same, how to maintain the identity information of the original face image? How do we apply face recognition on these obfuscated images?
- Face privacy protection is an import problem in automatic face recognition and analysis. However, the sensitive factors are determined by different applications, and should be relative variables. For example, if it is a face recognition application, then face identity is the target feature and is supposed to be maintained in the original face image; while other information, such as demographic attributes, are sensitive factors that should be removed. After privacy protection, we may not be able to obtain any other information, but we should still be able to identity people using the face images.
On the other hand, if the application is demographic attribute estimation, then demographic attributes, for example, age, gender, or race/ethnicity, are the key features that should remain in face images for estimation; while other information, like identity, health status, etc., ought to be treated as privacy information that need to be removed from original face images. After privacy protection, face recognition may fail to apply, whereas, the performance of demographic attribute estimation is expected to be no worse than the original face images.
Therefore, my question is, if we use DCC to obfuscate face images, making them hard to be recognized in both face verification, and demographic attribute estimation, what is the use case for these obfuscated face images? Are they still feasible for face recognition?
- The face recognition model is not state-of-the-art (SOTA). The current SOTA face model can achieve 99.8% accuracy on LFW. Why not use a SOTA method for evaluation?
Author Response
We would like to thank Reviewer 3 for the constructive comments and suggestions for improving the quality of the manuscript. We have carefully reviewed these comments point by point.

Round 2
Reviewer 1 Report
I have no other comments.
Author Response
Our paper has been further developed with your constructive comments. Thank you once again.